# HIV-1 diversity in viral reservoirs obtained from circulating T-cell subsets during early ART and beyond

Yuepeng Zhang[1][e], Fabian Otte[1][e], Marcel Stoeckle[2], Alexander Thielen[3], Martin Däumer[3], Rolf Kaiser[4,5], Katharina Kusejko[6], Karin J. Metzner[6,7], Thomas Klimkait[1]*, and the Swiss HIV Cohort Study

1 Molecular Virology, Department of Biomedicine, University of Basel, Basel, Switzerland, 2 Infectiology, University Hospital Basel, Basel, Switzerland, 3 Seq-IT GmbH & Co.KG, Kaiserslautern, Germany, 4 Institute of Virology, University of Cologne, Cologne, Germany, 5 German Center for Infection Research, Partner Site Bonn-Cologne, Cologne, Germany, 6 Department of Infectious Diseases and Hospital Epidemiology, University Hospital Zurich, University of Zurich, Zurich, Switzerland, 7 Institute of Medical Virology, University of Zurich, Zurich, Switzerland

e These authors contributed equally to this work.
* thomas.klimkait@unibas.ch

**Data Availability Statement:** SGA sequencing data have been deposited at GeneBank with accession numbers: PQ046328-PQ046399. NGS data have

## Abstract

Even during extended periods of effective immunological control, a substantial dynamic of the viral genome can be observed in different cellular compartments in HIV-1 positive individuals, indicating the persistence of active viral reservoirs. To obtain further insights, we studied changes in the proviral as well as in the viral HIV-1 envelope (Env) sequence along with transcriptional, translational and viral outgrowth activity as indicators for viral dynamics and genomic intactness. Our study identified distinct reservoir patterns that either represented highly sequence-diverse HIV-1 populations or only a single / few persisting virus variants. The single dominating variants were more often found in individuals starting ART during early infection phases, indicating that early treatment might limit reservoir diversification. At the same time, more sequence-diverse HIV reservoirs correlated with a poorer immune status, indicated by lower CD4 count, a higher number of regimen changes and more co-morbidities.

Furthermore, we noted that in T-cell populations in the peripheral blood, replication-competent HIV-1 is predominantly present in Lymph node homing TN (naïve) and TCM (central memory) T cells. Provirus genomes archived in TTM (transitional memory) and TEM (effector memory) T cells more frequently tended to carry inactivating mutations and, population-wise, possess changes in the genetic diversity.

These discriminating properties of the viral reservoir in T-cell subsets may have important implications for new early therapy strategies, underscoring the critical role of early therapy in preserving robust immune surveillance and constraining the viral reservoir.

also been deposited at GeneBank and can be accessed under PRJNA1153747.

**Funding:** YZ was financially supported by the China Scholarship Council (CSC201906170031). FO was paid through research funds of the Molecular Virology Laboratory, University of Basel, Basel, Switzerland. The work was supported by the University of Basel, Basel, Switzerland. The funders had no role in study design, data collection and analysis, decision to publish, or preparation of the manuscript.

**Competing interests:** K.J.M. has received travel grants and honoraria from Gilead Sciences, Roche Diagnostics, GlaxoSmithKline, Merck Sharp & Dohme, Bristol-Myers Squibb, ViiV and Abbott; and the University of Zurich received research grants from Gilead Science, Novartis, Roche, and Merck Sharp & Dohme for studies that Dr Metzner serves as principal investigator, and advisory board honoraria from Gilead Sciences and ViiV.

## Author summary

Our study demonstrates that a poor immune status associates with and may form a basis for HIV reservoir diversification. Reservoirs for replication-competent, infectious viruses are mainly found in TN and TCM, which harbor a small number of intact viral variants. A distinct population of archived HIV sequences, residing in TTM and TEM, is characterized by a shifted genetic diversity. While the former cells are likely to harbor a promptly re-activatable latent reservoir, the latter with often defective but more variable viral genomes could thus represent a source for retroviral recombination and reactivation.

## Introduction

It has been established that the primary HIV-1 infection event is typically initiated by a single or few founder viruses [1]. While a continuously growing population of infected cells rapidly emerges, the HIV-1 reservoir persists mainly as integrated proviral copies in various types of resting memory CD4+ T cells for decades and even during periods of suppressive antiretroviral therapy (ART) [2–5]. These resting memory CD4+ T cells are found to be either infected during a transient activation period or in the continuous resting state [6–8]. Main viral sanctuaries are localized to sites including lymphoid tissue, gut, and brain [9,10]. So far, neither ART nor various "shock and kill" strategies have been able to durably restrain the HIV-1 reservoir to undetectable levels [11–13], Thus proviral eradication is currently not possible with the rare exception of individuals after CCR5Δ32 allogeneic stem transplantation, where the invasive procedures bear high risks rendering procedures prohibitive for any broad application [14–16].

HIV-1 infection is often characterized by the rapid establishment of a genetically diverse viral population within months after primary infection, which is attributable to the high error rate of the reverse transcriptase (RT) enzyme [17], a rapid viral turnover [18], and the active immune selection processes in the host [19–23]. A longitudinal analysis of the viral sequence variation after seroconversion reveals that the mutational rate is approximately 1% per year until the sequence diversification may eventually plateau [24]. However, major gaps still exist in understanding the in vivo dynamics of viral diversification, evolution, and clinical latency during the early period of antiretroviral therapy (ART).

Various approaches have been developed to quantify and track the HIV-1 diversity under drug selection pressure, e.g. by sequencing the envelope V3 region of plasma virus or in cells of individuals receiving ART [25,26]. The highly variable V3 loop is thought to serve as a suitable indicator for viral variability and, at the same time, as a determinant of the viral tropism for cells displaying the chemokine receptors CCR5(R5) or CXCR4(X4) [27].

For the clinical application major conclusions regarding therapy decision are typically simplified to the determination of viral load, CD4 levels and inflammation markers. However, our work supports the utility of additional markers including the quantitative determination and analysis of intracellular viral poly-A RNA (mRNA) and of quantitative levels of the common non-pathogenic bystander virus Torque teno virus (TTV).

Recent work established that the viral load of the TTV may serve as a predictive marker in the context of the immunological competence of a given person [28,29]. As HIV infection and persistence are closely linked to the immunological capacity, we included the determination of TTV levels in this study.

Our study focused on the detailed molecular examination of the viral reservoir during the critical period between therapy initiation and full viral suppression. Hence, we chose specimens from individuals recently diagnosed with HIV-1.

Using Next Generation Sequencing (NGS) this study aimed at either identifying highly diverse virus reservoirs or reservoirs with a single dominating virus variant over the monitoring period. Individuals with highly diverse reservoirs were defined as those who exhibited a continuously changing viral reservoir >3 weeks after starting ART, characterized by more than 3 variants, each with a relative abundance greater than 5%, identified by NGS. High diversity was generally linked to poorer immune function and chronic/late clinical presentation. Our study demonstrates that replication-competent viruses were only found in less differentiated cells, i.e TN and TCM, although viral transcripts were detectable in all T-cell memory compartments. In contrast to TN and TCM, TTM and TEM contained much more shifted and distinctive viral repertoires.

## Results

### Study participants

To address principles of viral evolution and genetic diversity in the HIV-1 reservoir throughout infection, we focused on the persistence of viral gene activity during different phases of HIV infection. This study focused on individuals shortly after a clinically recorded first positive HIV test (Fiebig stages are not recorded in the patient files) to cover on the one hand periods immediately after a primary HIV infection and on the other phases of an untreated HIV infection in late presenters (see S1 Table). Among the 9 persons with HIV (PWH) in this study, three individuals (P6, P7, P8) were confirmed to be in the primary phase of infection and two (P3, P5) with a confirmed chronic infection. For subjects P1, P2 and P4 no recency date was available to define a likely time point of infection. Individual P4 had a fully suppressed viral load already at the treatment start, suggestive of a prior therapy, which the person had not disclosed to the treatment center for fear of being rejected (S1 Table).

As a chronic virally suppressed control with a documented earlier acute infection phase, we included P9 and monitored from day 531 after treatment start (S1 Table). For this individual plasma viral loads (VL) were below 100 copies/ml at all analyzed time points (between days 531 and 1454 after ART start).

CD4+ T cell counts continuously increased during the study period in 8 of the 9 individuals, (Fig 1A). For individual P3, even during the extended period of fully suppressed viral loads (VL) and during the entire study period CD4+ cells did not recover to levels above 200 cells/μL. In 4/9 individuals (P3, P5, P7, P8), viral loads became undetectable within less than 55 days after treatment initiation (from 5,972–43,064 copies/mL). P4 and P9 presented with a suppressed VL already at enrolment. The other three individuals (P1, P2, P6) remained viremic for 168–386 days after therapy start but suppressed viral replication thereafter. One single viral blip to <250 c/mL during the study period was noted for P2.

Regarding relevant comorbidities, individuals P3, P4, and P5 were co-infected with HBV, P6 presented after HCV therapy, P1, P4, and P9 had a history of mycobacterial infections, P8 had a Lues history with appropriate medication, and all study participants were sero-reactive for CMV. In addition, individual P5 presented with severe wasting syndrome, and P6 was diagnosed with prostate cancer (S2 Table).

### Reservoir dynamics before and during therapy

To assess viral dynamics and intracellular activity, proviral DNA loads and cell-associated poly-adenylated (pA) RNA loads were determined at the earliest time point available at or shortly after diagnosis and at each follow-up visit (Fig 1B). DNA loads stayed at stable levels in all individuals with between 3 and 4,144 copies/$10^6$ cells. Correspondingly, intracellular pA RNA was detectable in 4/9 individuals (P2, P3, P4, and P6; Fig 1B), ranging from 19 to 3,864

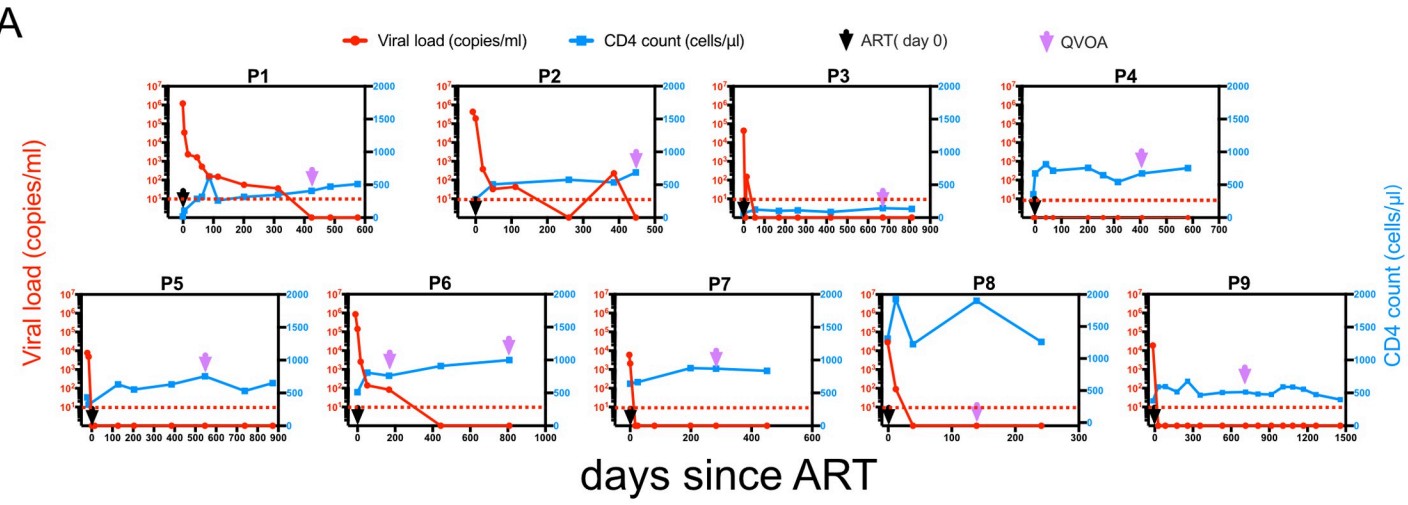

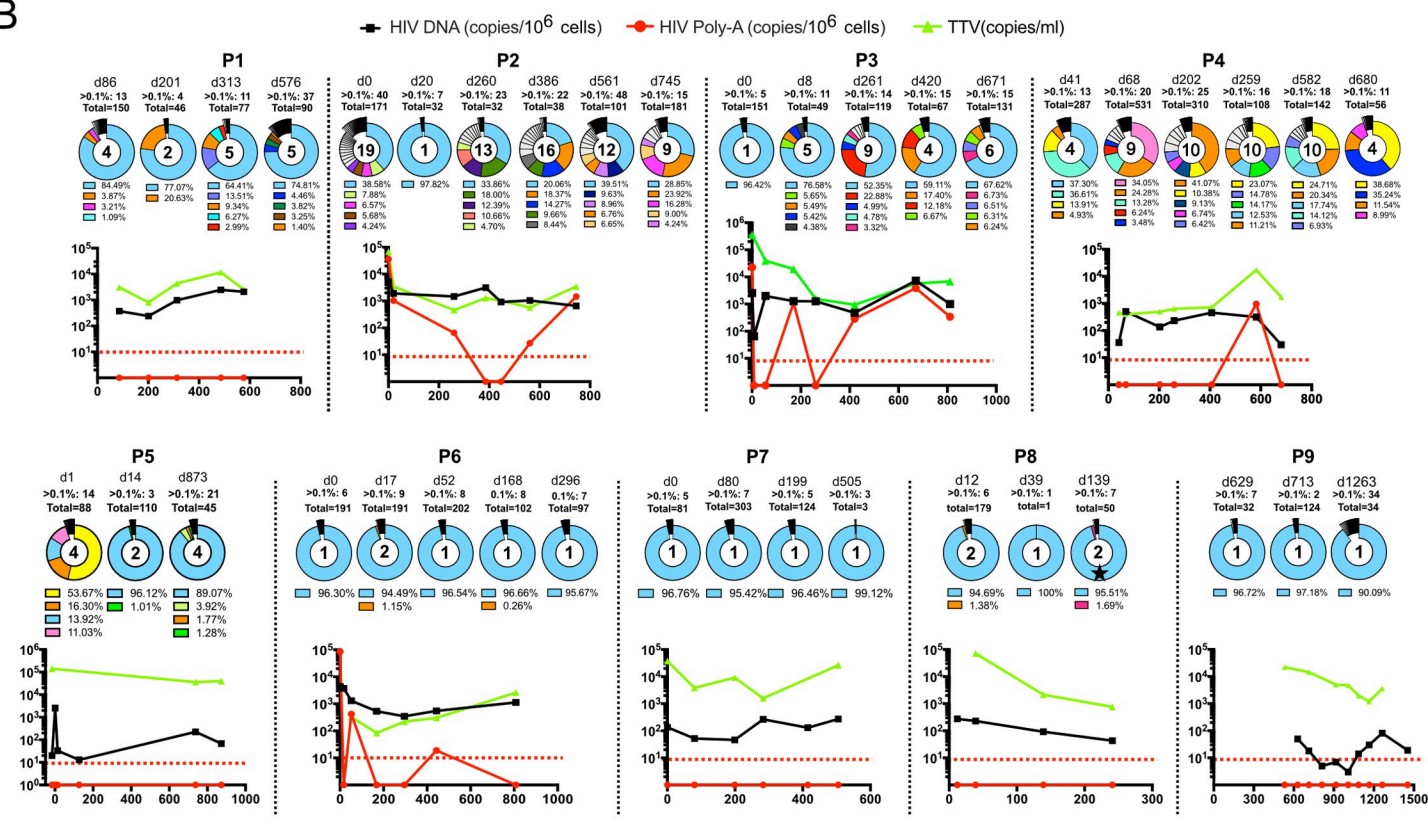

**Fig 1. Longitudinal characterization of 9 HIV-1-infected individuals followed from the day of diagnosis.** (A) Longitudinal HIV-1 viral loads and CD4+ T cell counts. Black arrows denote the start time of ART treatment, purple arrows highlight time points for Viral outgrowth (VOA) sampling, and the red dotted line marks the limit of detection/full viral suppression. Viral load (RNA copies/ml) in red left y-axis, CD4+ T cell counts (cells/μL) in blue right y-axis. (B) Dynamics of reservoir diversity over time. Pie charts show HIV-1 proviral V3 loop diversity obtained by NGS. Respective sampling time points are shown above each pie chart in days after therapy with number of variants above 0.1% relative abundance. Variants with a proportion below 1% are jointly summarized in black and only the top 5 variants are shown in color with respective abundances shown in the legend below the pie chart. Each color represents one separate virus variant. The number inside the pie charts highlights how many sequences have a relative abundance of >1%. For individual P8, one conservative amino acid change in the V3 loop of the dominant variant on day 139, is indicated by an asterisk. In scatterplots, a black line denotes HIV-1 proviral DNA load, red lines reflect HIV-1 pA loads, a green line depicts TTV load, and red arrows mark the limit of HIV-1 pA detection. The red dotted line indicates the detection threshold (<10 copies). Each individual ID is stated above respective plots.

copies/$10^6$ PBMCs. Intracellular viral transcription remained detectable for long periods (at least 745, 582, 811 or 443 days after treatment for each respective individuals, respectively). This is suggestive of continuous viral transcriptional activity in the viral reservoirs despite the absence of any detectable particle production or any release into the peripheral blood, where viral loads remained undetectable.

Only a small fraction of viruses found in circulating blood directly emerge from persisting reservoirs. Earlier studies suggested that the main contributors to the proviral reservoir in the periphery are short-lived productively infected cells, whereas the persisting reservoir is assumed to be rather small at this point in time [30,31].

To understand the viral dynamics during the early course of therapy, we focused on changes in indicative Env V3 loop sequences of the proviral DNA and analyzed them by NGS over time under treatment. We found mainly two patterns: in four individuals (P1, P2, P3, P4) viral V3 signatures revealed significant changes over time, while the other 5 individuals (P5, P6, P7, P8, P9) showed a rather stable proviral sequence over time where one single variant represented 95–97% of the entire proviral pool of HIV-1 genomes (Fig 1B). Also, in individuals P1, P2, and P3, the analysis identified one predominating HIV-1 variant for each time point, which reflected 33.9–97.8% of the whole proviral population. However, these virus variants were consistently accompanied by several additional variants at varying frequencies between 1% and 23.9% of the entire proviral population. For individual P4, the composition of HIV-1 variants remained highly dynamic at all time points. This individual had switched therapy several times during the study period (S2 Table). These proviral V3 data were further complemented and validated by an NGS analysis of V3 regions from an analysis of virus-derived RNA during periods of unsuppressed VL (S1 Fig). For individual P5, although presenting with 4 major proviral variants at baseline (S3 Table), only two of these variants were detectable in plasma, and one of them accounted for 95.9% of the entire virus population (S1 Fig). Although in P2, plasma virus was fully suppressed by day 477 of therapy, intracellular HIV transcripts remained detectable at days 561 and 745 (Fig 1A and 1B). And after VL suppression, the latent reservoir became successively dominated by one single virus variant (Fig 1B).

In individual P8, one dominant proviral variant was identified on days 12 and 39 days after treatment initiation. On day 139, the same sequence with only a single amino acid change in V3 was recovered (Fig 1B).

In most individuals (5/9) a CCR5 tropism predominated in the proviral population throughout the study period. Only three individuals, P1, P5 and P9, had exclusively X4 tropic virus sequences at all time points. In P2, the proviral virus population possessed mixed tropism and changed with a gradual increase of R5-tropic variants (S2B Fig). P4, however, showed a reversed trend with X4-tropic variants increasing during follow-up (S2D Fig). The dominating CCR5 tropism reflects the clinical course of our cohort and is typically seen when immune functions recover as indicated by rising CD4 counts [32,33].

## Genetic diversity of HIV-1 in memory T cell subsets

While numerous studies have established that CD4+ T lymphocytes harbor the majority of the HIV reservoir, the dynamics and evolution inside individual T-cell subsets during early therapy periods are still largely unclear. Therefore, we investigated if representative aspects of the HIV-1 reservoir diversity could be deduced from cells that transiently occur in circulating blood. For this purpose, CD4+ T cells were harvested from a larger sample of native peripheral blood and sorted into TN, TCM, TTM and TEM CD4 memory T cells, while one aliquot of unsorted CD4-positive cells served as a basic comparator to the various memory compartments. Proviral NGS analysis showed that in most individuals the repertoire of identified

variants in the TN and TCM compartments was largely similar to that identified for the bulk CD4+ T cells or for earlier time points (Figs 1B and 2A). For P1, we detected in TTMs a predominant proviral variant (90.4% of the whole proviral population) that was only found as a small proviral minority in bulk CD4, TN, or TCM (Fig 2A). This small minority was already present for bulk proviral NGS sequencing during earlier treatment periods, e.g. on days 86 and 201 after therapy (Fig 1B). The same phenomenon of proportionately higher predominating variants was seen in the TTMs of P2, P3, P4, and P7, and in the TEMs obtained from P1, P2, P4, and P5 (Fig 2A).

The cells from individual P8 consistently contained a very low proviral load, and even after 5 days post-stimulation an amplification of the env V3 region remained unsuccessful (S3 Fig).

## The competence of T-cell subsets to reactivate infectious virus

Most integrated proviruses persist for extended times as replication-defective viral genomes in the affected individual [34,35]. Thus, inferring viral dynamics just from the proviral load can lead to misrepresentation. We therefore combined sequencing, transcriptional, translational, and functional techniques to allow a predictive interpretation.

The in vitro stimulation of T-cells from 3/9 individuals (P2, P3, P6) induced high transcriptional HIV activity in all memory subsets. Stimulated cells from P1, P7 and P8 yielded pA RNA synthesis only in TN and TCM. In the T-cell populations from 3/9 individuals (P4, P5, P9) no viral transcription could be induced (S3A Fig) and overall, levels of proviral HIV-1 DNA as well as viral transcripts were similar in every T cell subset (S3B Fig).

Infectious virus was identified only in TN and TCM of P1 and in the TN of P7 by GERDA [36] and by a viral outgrowth assay (VOA) using an LTR-driven reporter system [37] (S4A and S4B Fig). For individual P1 we determined 0.64 infectious units per million (IUPM) in TN cells (confidence interval {0.31, 1.35}) [38] and 0.74 IUPM in TCM (confidence interval {0.19, 2.98}). The subsequent NGS analysis found in the X-Gal positive outgrowth wells (of TN and TCM) two distinct virus variants (Fig 2B, blue and orange). These two variants had already been detected by NGS at earlier time points as the most abundant proviral species (Fig 1B) and were also identified by NGS as dominating proviral variants in the less differentiated TN and TCM memory compartments (Fig 2A). To further examine the genomic composition of the reactivated viruses, single-genome amplification (SGA) was performed of the 3' half of the viral genomic RNA. For P1, two wells (no. 1 and 3) with different V3 loop sequences in the population of TN cells, and well no. 3 of the TCM cell cultures were picked to represent relevant cultures for SGA (Fig 2B). Each well could be discerned as one cluster in the phylogenetic tree. In S5A Fig, the first 1000 bp of the red and blue clades are almost identical, suggestive of a potential viral recombination event. All sequences share the same V3 region implying a common ancestral virus that has further evolved inside the host through mechanisms like recombination events, RT- or APOBEC-induced mutations.

The analysis of the X-Gal positive TN culture supernatant of individual P7 accounted for 0.08 IUPM {0.01, 0.60}, and detected a single virus variant by SGA (S5B Fig). This was again in full agreement with the results of the NGS on proviral and viral variants at earlier time points (Figs 1B, 2C and S2F).

No infectious virus could be rescued from samples of the other examined individuals (S4C–S4F Fig) and no viral RNA was detected in the supernatant of stimulated cell cultures that would indicate the presence of intact virus.

NGS and SGA results excellently support the data obtained with VOA and GERDA. They further show that those variants, most abundant in TN and TCM cells, reflect the replication-

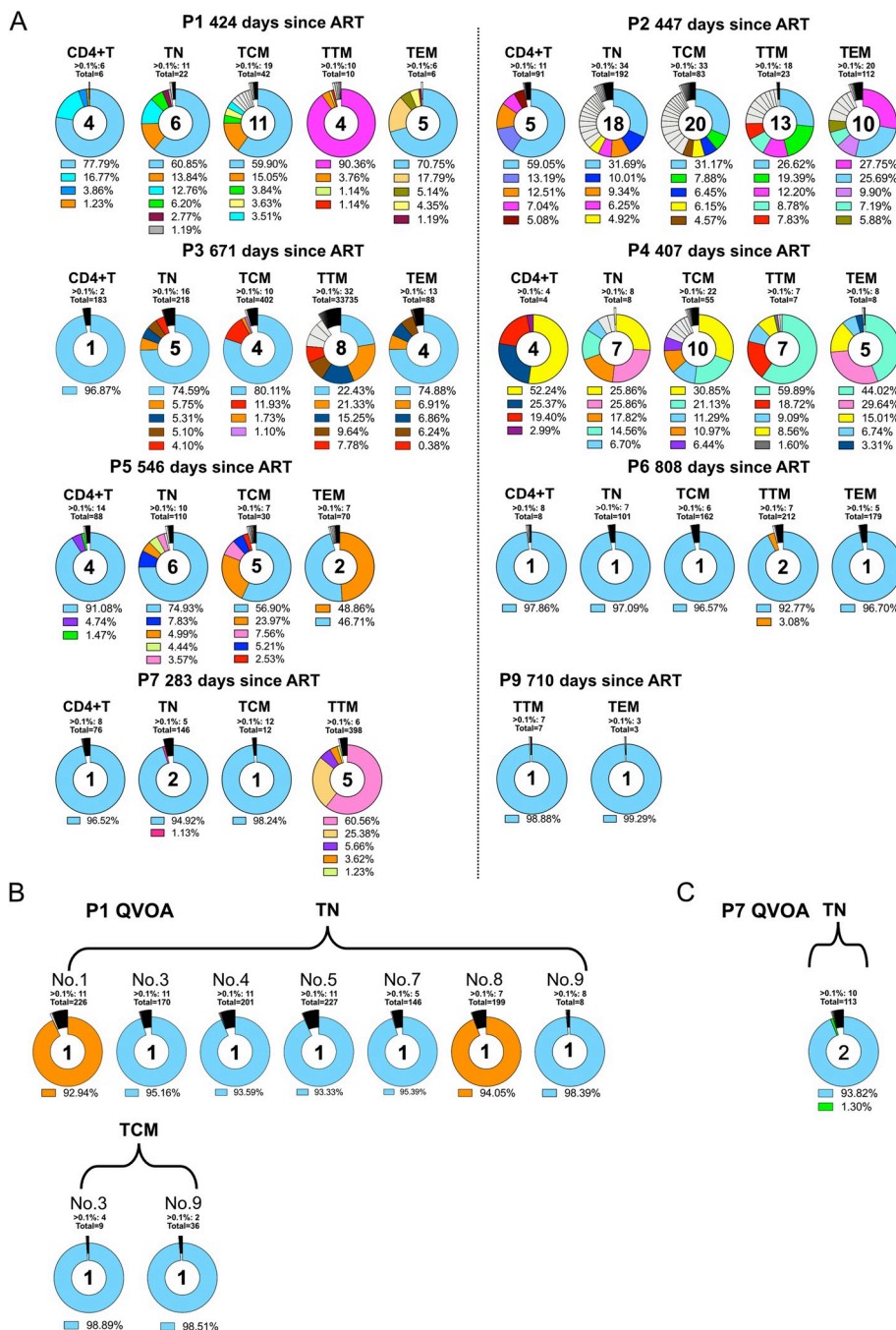

**Fig 2. Dynamics of the HIV-1 reservoir in memory T cell subsets.** Proviral V3 loop diversity, determined by NGS 5 days post-stimulation, represented by pie charts for bulk CD4 T cells and each memory T cell subset. (A) Variants with a proportion below 1% were separated and only the top 5 variants are shown in color with respective abundances shown in the legend below the pie chart. Each color represents one separate virus variant. The number inside the pie charts highlights how many sequences have a relative abundance of >1%. Each participants ID is stated above each plot. B,C: NGS V3 loop diversity of viral RNA from viral outgrowth positive wells. (B) Reactivated virus of TN and TCM form P1, (C): Reactivated virus of TN form P7.

competent viral reservoir (Figs 1B, 2 and S4). In addition, they indicate that minor virus variants that had been identified early after the initial infection event can persist over long periods.

### Immune competence in HIV-1 infected individuals

To investigate aspects of the HIV-induced dampening of crucial immune functions during early periods after diagnosis, we compared CD4+ and CD8+ T cell populations in individuals with high variability to those with only a few viral variants in the circulating blood. In 7 individuals, the CD4 count was within 18-999/μL; in P3, the CD4 count was between 65-144/μL; in P8, the CD4 count consistently ranged from 1,310-1,928/μL. Although lymphocyte, CD4 and CD8 counts were not significantly different among the two groups, CD4 counts show a trend of higher CD4 levels for all chosen time intervals in the low diversity group compared to the high diversity group (Fig 3A–3C). Interestingly, the CD4 gain of the low viral diversity group was highly significant in the first year of ART.

Searches for biomarkers and observations of bystander infections have revealed a link to other non-pathogenic viruses such as the TTV. We set out to analyze plasma viral load levels of the highly prevalent TTV, which has no link to disease exacerbation or AIDS, for further verification of immune competence. The TTV VL has been reported by others to inversely correlate with the immune status, e.g. in the context of stem cell transplantation but also during the clinical course of an HIV-1 infection [28,29,39,40]. All 9 participants of this study tested positive for TTV at all time points between 0 to 1,263 days after therapy initiation where a sample was available, with TTV viral loads ranging from 84 to 358,737 copies/mL (Fig 1B). With the onset of ART, TTV loads decreased in the group of individuals presenting with a high genetic HIV diversity in the first year and stayed stable in the following years, while in the low diversity group, TTV loads remained stable independently of the individual treatment situation.

Poly-A quantification revealed falling levels of transcripts over time in both groups with faster drops in the low diversity group. However, due to undetectable poly-A levels in some of the individuals no statistical test could be performed.

## Discussion

Understanding the dynamics and persistence of HIV-1 reservoirs during periods of suppressive ART is crucial for optimizing the efficacy of therapeutic interventions that pursue to cure HIV-1 infection. This study focused on the viral diversity in key cellular compartments that are part of or may transiently appear in the peripheral blood. The work assessed changes from immediately after HIV diagnosis to the early periods of successful therapy in acute and chronically infected individuals. Recent findings suggest that during these periods, viral reservoirs of HIV-1 are neither "stable" nor "silent" but often retain a certain transcriptional activity, which may indicate a dynamic evolution, mainly driven by the immune pressure of the host [41].

While total HIV-1 DNA, plasma viral load, and CD4 cell count are independent predictors of viral rebound and disease progression [42–45], these markers are not similarly applicable to situations during suppressive ART: viral replication is fully controlled, CD4 cells recover, and yet, copies of intact or replication-defective viral DNA are retained and persist [46,41]. This study therefore focused on intracellular poly-A transcripts of HIV-1 as a more predictive expression marker than the sole presence of sequences of proviral DNA. In combination with the assessment of two discrete translational events (Gag and Env), the transcriptional component adds an important element for determining the replication competence of a persisting virus and its contribution to pathogenesis [36,47,48].

Our study confirms that high expression levels of HIV-1 poly-A transcripts are consistently observed in PBMCs and peripheral memory T cell subsets even during continuous, suppressive ART. In turn, it is conceivable that it may be the very immunological conditions of severe disease-providing situations that favor an HIV re-activation or stabilize viral reservoirs in vivo [47–50].

As an important technical detail, we noted for several individuals that, even during periods of detectable HIV-1 transcription, a successful viral outgrowth from in vitro-activated PBMCs was reached only after repeated in vitro cell stimulation. This phenomenon has also been reported by others in the field [36,50,51].

Although our study encompasses quite heterogeneous clinical profiles as they are typical for the point of HIV diagnosis, it identified mostly two principal situations: one, where the HIV-1 reservoir consisted mainly of one unique or highly predominant variant, which remained prominent over the entire study period; and another one, in which the reservoir consisted of a swarm of co-existing, genetically diverse HIV-1 variants. This latter pattern tended to stay very variable over time with waxing and waning variants.

Situations with one predominating and stable virus strain could resemble the most successful viral offspring after the initial HIV-1 infection event, either co-transmitted as a minor variant or evolved from the founder virus(es) [52]. It has been reported that during the acute HIV-1 infection phase, only very few virus variants are found to circulate in the blood. Also after therapy initiation during the primary/very early infection phase, only small HIV-1 reservoirs have been observed, which may limit residual virus replication and spread to body compartments [53]. These observations provide a very plausible explanation as 4/5 individuals of the low diversity group were diagnosed during the early infection phase. It is tempting to speculate that the situation of P5 might also fall into the early infection phase.

The second group of individuals in our study, in whom the HIV-1 reservoir is characterized by a continuously high viral diversity during therapy has also been reported by others [54,55]. Of note, these individuals tended to present a higher virus variability also in the plasma at baseline or at early timepoint after treatment (P2 as an example). In these individuals, dynamic changes of the proviral representatives were repeatedly observed in the T-cell reservoir under ART. This situation was typically accompanied by lower CD4 cells and by higher TTV loads (Figs 1B, 2A, 3B and 3D). Although differences for CD4 between both groups were not (yet) significant, there is a clear trend visible for all selected time intervals. This trend is also seen for TTV in the high-diversity group from before therapy to the first year under ART.

Earlier studies have suggested that, although levels of HIV-1 integration are lower in TNs compared to other memory T cells, they serve as main contributors to the reservoir of intact proviruses [56–58] and possess a longer half-life than other memory T cells [59,60]. Proviral sequences identified in naïve T cells tend to be distinct from those of more differentiated memory T cells, and the number of clonal HIV sequences increases during differentiation into an effector memory phenotype [56].

TCM cell populations are characterized by minimal cell proliferation. Their inherent ability to persist for very long periods is suggestive of a role as a stable long-term reservoir for HIV-1 [5,61,62]. Both aspects are in excellent agreement with the observation of this study that TN and TCM represent the key reservoirs for replication-competent and infectious viruses, and both mostly harbor single intact viral variants [5,56–58].

Our conclusion that only a smaller contribution of archived HIV-1 sequences comes from TTM and TEM, in which a shifted variability characterizes the viral sequences, would fit with their physiological role. TTM and TEM are considered to serve as the "patrolling" memory subsets that traffic to sites of pathogenic encounters [63,64]. We speculate that specifically the TTM and TEM are repeatedly exposed to virus sequences within strictly tissue-resident reservoirs as those compartments that the rather lymph node-resident TN or TCM cells never encounter.

The observed two principal viral expression- and persistence patterns may indicate the existence of two principal viral reservoirs in distinct T-cell compartments [5]. While TN and TCM could mainly serve as HIV-1 reservoirs for maintaining genetically stable, functional viral

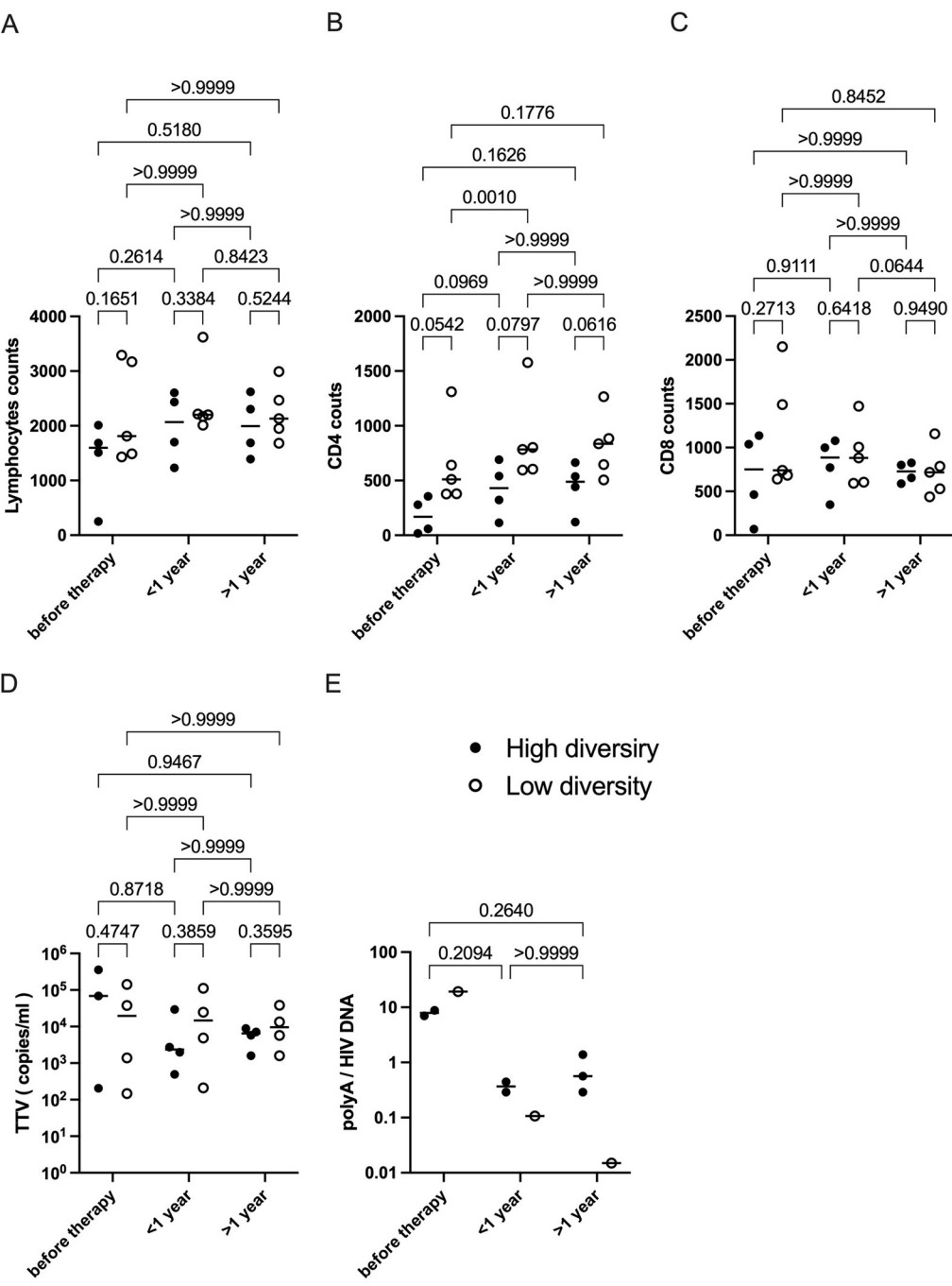

**Fig 3. HIV-1 viral dynamics and immune competence.** (A-E) Immunological comparison of the 'high diversity group' and the 'low diversity group'. (A) Lymphocyte count. (B) CD4 count. (C) CD8 count. (D) TTV load. (E) HIV polyA / HIV DNA. Averaged values were plotted for each individual and time interval. Black line denotes median value. Two-way ANOVA was performed with Geisser-Greenhouse correction and Bonferroni's multiple comparison test.

variants, a dynamic pool of highly variable and partly defective variants, resides in TTM and TEM. We note that the latter T-cell subsets harbor different patterns and/or prevalence of genetic HIV-1 variants. This may favor viral recombination events that are different from those in TN or TCM. It is therefore tempting to speculate that the shifted genetic diversity of

proviral genomes in these cell types could serve as a starting point for viral genome recombination and the re-emergence of new replication-competent virus variants [65,66].

A clear limitation of our study is that the chosen study group represents only a small set of individual infection situations with no stringent definition of an early HIV infection. Nevertheless, all have in common that therapy was initiated very soon after diagnosis. In this respect, an important common feature is pictured in the study. Moreover, we note that a larger cohort size will be needed to verify any "general mechanism", since each viral reservoir naturally has its own dynamics and may be shaped by treatment and regimen as well as exogenous factors (comorbidities, episodes with unrecognized blips and viral replication, etc.). Besides using a larger number of samples, the analysis would principally benefit from using higher blood volumes to increase the chances of reactivating dormant proviruses. Yet, the Swiss cohort ethical approval only allows full blood volumes of up to 40 mL. Irrespectively, TCM (mean 12.30%), TTM (mean 8.49%) and TEM (mean 20.53%) are similarly represented with equal normalized proviral and transcriptional loads (Figs S3B and S6). Furthermore, our sequencing analysis mainly focused on the variable V3 Env region as it has been reported to be under continuous rearrangement, which can lead to events such as genomic mutations that can escape V3-targeting antibodies. We therefore interpret the observed enhanced dynamics rather as an advantage for emerging viruses. The NGS data are further supported by SGA sequencing data that analyses the entire 3' half of the HIV-1 genome. At the same time, a continuing intra-host evolution under (fully suppressive) therapy is not expected in the peripheral blood. To further follow the long-term stability of viral genomes or an increase of mutations over time studies covering longer monitoring periods will be needed.

In conclusion, this study demonstrates and confirms other reports that key reservoirs for replication-competent HIV-1 predominantly reside in less differentiated TN and TCM memory T cell subsets, consistently harboring a rather homogenous population of intact virus genomes. In parallel, the archived proviral reservoir localizing to TTM and TEM contains mostly inactive viruses with shifted genetic diversity.

Our results provide strong evidence for an inverse correlation between immune competence and the genetic diversity of the viral reservoir, supporting the importance of early therapy to maintain maximal immune surveillance and to restrict the viral reservoir. Follow-up studies should include bigger cohorts to study further the combination of HIV-1 transcription levels and plasma TTV loads to provide potentially useful indicators for obscure HIV activity in viral reservoirs.

Further studies will be needed to investigate whether an interference with the proliferation and differentiation of different T cell subpopulations will impact the clearance of critical HIV-1 reservoirs.

## Methods

### Ethics statement

Approval for this study, SHCS project #839, was obtained from the local ethics committees of all study sites of the Swiss HIV Cohort Study (Kantonale Ethikkommission Bern, Ethikkommission beider Basel, Ethikkommission des Kantons St. Gallen, Kantonale Ethikkommission Zurich, Comitato etico cantonale del Ticino, Commission d'ethique de la recherche linique de la faculte de biologie et de medicine de l'universite de Lausanne and Comite d'ethique du department de medicine des hopitaux universitaires de Geneve), and written consent from all participating individuals is available. All infection recency information was obtained through the clinical records available at the SHCS centers.

## Cells

HUT4-3 cells are produced from HUT78 cells (M. Martin, NIAID, Bethesda, MD, USA), which constitutively express a subtype B isolate of HIV-1 [67]. SupT1 cells were from the laboratory of M. Martin. TZM-bl cells were obtained through the NIBSC center for AIDS reagents [68], and Ramos cells were obtained from Dr. Gertrud Steger (University of Cologne, Cologne, D).

## Cell culture

HUT4-3 cells, Ramos cells, and SupT1 cells were maintained in RPMI 1640 medium supplement with 10% FBS, penicillin (100 U/mL), and streptomycin (100 U/mL) (Gibco/Thermo Fisher, Allschwil, CH). TZM-bl and SXR-5 cells were maintained in DMEM medium supplement with 10% FBS, penicillin (100U/ml), and streptomycin (100U/ml). Cultures were maintained in 5% $CO_2$ at 37˚C.

## PBMC isolation

Up to 40 mL of EDTA-blood were obtained from consenting, recently diagnosed HIV-1 positive individuals. PBMC were isolated by density gradient centrifugation on the day of harvest and were either stained directly or cryo-preserved in FBS with 10% DMSO for later time analysis.

## HIV-1 DNA quantification

The proviral load in cells was quantified as described by others [69]. In brief, a single-plex qPCR was done using primer and probe pairs for either HIV-1 LTR or CCR5 [69]. For standards, HUT4-3 genomic DNA was used (2 HIV-1 copies per cell) and HIV-1 DNA extracted by Maxwell RSC Cultured Cells DNA Kit (Promega, Madison, USA). For quantification, the GoTaq Probe qPCR Master Mix (Promega, Madison, USA) was used, PCR conditions were as follows: initial denaturation for one minute at 95˚C, followed by 45 cycles of alternating 15 seconds at 95˚C and one minute at 60˚C. Samples were run on a 7500 Fast Real-time PCR System (Thermo Fisher, Walthan, MA, USA).

## HIV-1 pA RNA transcript quantification

Intracellular poly-A HIV-1 transcripts were quantified according to a recent publication [70]. Briefly, intracellular RNA from fresh or cryo-preserved PBMC were isolated with Promega Maxwell RSC simply RNA cells kit (Promega, Madison, USA), and eluates were immediately frozen at -80˚C. HIV-1 poly-A transcripts were quantified by qRT-PCR Brilliant III probe master mix kit (Agilent, Santa Clara, USA) using the titrated inhouse plasmid DNA pMVQA (Addgene ID 197894) as standard. Viral transcripts were normalized to cell count using CCR5 as described elsewhere [69]. qPCR was performed on a 7500 Fast Real-time PCR System (Thermo Fisher, Walthan, MA, USA) using the following cycling conditions: initial 50˚C for 10 min and 95˚C for 3 min, followed by 45 cycles of 95˚C 15 sec, 60˚C 30 sec.

## TTV DNA quantification

Viral DNA was extracted from serum samples by the Maxwell RSC Plasma viral TNA Kit (Promega, Madison, WI, USA). The TTV DNA was quantified by real-time PCR as previously described [71]. qPCR was performed using the GoTaq Probe qPCR Master Mix 2X Kit with the following primer and probe set: AMTS 5'-GTGCCGIAGGTGAGTTTA-3' and primer AMTAS 5'-AGCCCGGCCAGTCC-3', probe AMTPTU 5'-FAM-TCAAGGGGCAATTCGG GCT-TAMRA-3', qPCR was performed using the 7500 Fast Real-time PCR System (Themo

Fisher) with the following cycling conditions: initial 95°C for 10 min, followed by 45 cycles of 95°C 15 sec, 58°C 60 sec.

## FACS staining

PBMCs were washed at 37°C during all PBS washing steps. FcR was blocked for 5–10 min at 37°C, followed by 15 min Live/Dead and CCR7 staining at 37°C. Thereafter, residual surface markers were incubated for an additional 20 min at RT. For all subsequent washing steps cells were washed in staining buffer (PBS, 2 mM EDTA, 1% human serum). For intracellular staining, cells were fixed for 30 min at RT in PBS 2% PFA. Fixed cells were then permeabilized twice with ICS Permeabilization Wash buffer (BioLegend) prior to intracellular staining. Stained cells were washed twice in the perm-wash buffer before acquisition. For FACS following antibody panel was used: CD3 (BV510) (catalog no. 300447, BioLegend), CD4 (BV650) (catalog no. 300535, BioLegend), CD8(A700) (catalog no. 301027, BioLegend), CD14/CD19 (APC-Cy7) (catalog no. 301819/302217, BioLegend), CD25(BV605) (catalog no. 302631, BioLegend), CD28 (PE-Dazzle) (catalog no. 302941, BioLegend), CD45RA (BV711) (catalog no. 304137, BioLegend), CCR6 (PerCP-Cy5.5) (catalog no. 353405, BioLegend), CCR7 (APC) (catalog no. 353213, BioLegend), PD1 (BV421) (catalog no. 329919, BioLegend), HIV Gag (PE) (catalog no. 6604667, Beckman Coulter). For GERDA a biotinylated bnAb mix was used (3BNC117, 10–1074, PG16 all provided by F. Klein, Cologne, GER). bnAbs were subsequently bound by an Anti-Biotin-VioBright 515 (Miltenyi) secondary antibody. Except for HIV Gag (KC57, Beckman Coulter) all antibodies were purchased from BioLegend. An LSR Fortessa (BD) was used for cell analysis.

## Sorting

Up to 40 mL of whole blood was drawn from HIV-1 positive individuals on ART, PBMC were isolated by density gradient centrifugation using Histopaque 1.077 g/mol (Sigma). Subsequently, untouched CD4+ T cells were isolated by negative selection (CD4 T Cell Isolation Kit, Miltenyi), The individual T cell memory compartments of CD4 +T cells were separated by MACS using CD45RA microbeads (CD45RA MicroBeads human, Miltenyi), CD62L microbeads (CD62L MicroBeads human, Miltenyi), CD27 microbeads (CD27 MicroBeads human, Miltenyi) to separate T naïve (TN: CD45RA+CD62L+CD27+), T central memory (TCM: CD45RA-CD62L+CD27+), T translational memory (TTM: CD45RA-CD62L-CD27+)), and T effector memory (TEM: CD45RA-CD62L-CD27-) cells.

## Viral outgrowth

Sorted cell fractions were plated in 96-well plates. Per well 1-3X10E5 of sorted cells were plated with 10E4 SupT1 hu R5 as amplifier cells in TexMACS Medium (TexMACS Medium, Miltenyi), 100 IU/mL IL-2 and anti-human CD3/CD28/CD2 Antibody stimulation cocktail (Stem Cell Technologies). Every 2–3 days, medium was removed and replaced with fresh medium containing 50 IU IL-2. On days 7 and 14 cells were re-stimulated with antibody cocktail. From the 7th day onward, viral outgrowth was determined by GERDA and by X-Gal stain to follow viral propagation over time.

## X-Gal reporter assay

To monitor fully infectious viral particle release, infections were visualized by an HIV-1-induction system based on an integrated LTR-lacZ gene in the stable reporter line termed SXR5 [37]. 2-3X10E6 SXR5 (or TZM-bl) cells were plated in 24/48-well plates one day before

co-culture with putatively infected T cell subsets. Co-cultures were incubated for 2 days. Cells were finally fixed by 2% formaldehyde and infection events were visualized by X-Gal staining, for microscopic imaging, we used the Leica DMi1 system.

## FACS analysis

Flow cytometry data were analyzed using FlowJo v10.9.0 software (TreeStar).

## Single genome amplification

Viral RNA was extracted from culture supernatant using Maxwell RSC miRNA Plasma and Serum Kit (Promega, Madison, WI, USA) with an additional DNaseI digestion step. SGA of 3' half genome sequences was performed as previously described [72]. In brief, cDNA was synthesized using the SuperScript IV First-strand Synthesis System (Invitrogen, Carlsbad, CA) with reverse primer R-9632 5'-ACTACTTGAAGCACTCAAGGCAAGCTTTATTG-3'. cDNA was then serially diluted until only 30% of PCR had positive amplification using a nested PCR with Herculase II Fusion DNA Polymerase (Agilent, California, USA). This dilution step most likely contains amplicons derived from a single cDNA molecule. The outer PCR included sense primer F-4875 5'-CAAATTAYAAAAATTCAAAATTTTCGGGTTTATTACAG-3' and antisense primer R-9626 5'- TGAAGCACTCAAGGCAAGCTTTATTGAGGC-3' using PCR conditions as follows: 94˚C for 2 min followed by 35 cycles of 94˚C 15 sec denaturation, 58˚C 30 sec annealing and 68˚C 4 min extension, with a final extension of 10 min at 68˚C. The nested PCR was done with sense primer F-4900 5'-GGGTTTATTACAGGGACAGCAGAG-3' and antisense primer R-9602 5'-TGAGGCTTAAGCAGTGGGTTCC-3'. PCR conditions were: 94˚C for 2min followed by 45 cycles of 94˚C 15 s denaturation, 58˚C 30 s annealing, and 68˚C 4 min extension, with a final extension for 10 min at 68˚C. All PCR products were verified on a 1% agarose gel before sequencing.

## V3 amplification for next generation sequencing

For nested PCR amplification, the Herculase II Fusion DNA Polymerase Kit (Agilent Technologies) was used. 1 μL of extracted gDNA or purified first PCR product was mixed with 24 μL of Master Mix. To normalize for PCR amplification bias each PCR was done in triplicates. For the outer PCR primer F-6553 5'- ATGGGATCAAAGCCTAAAGCCATGTG -3´ and R-7801 5´- AGTGCTTCCTGCTGCTCCCAAGAACCCAAG -3' were used. Cycling conditions were as follows: initial three minutes at 95˚C, 30 cycles of denaturation for 15 seconds at 95˚C, annealing for 20 seconds at 60˚C and extension for 45 seconds at 72˚C. The final extension was done for three minutes at 72˚C. The second PCR used primer F-6848 (5´- AGGCCTGT CCAAAGGTATCCTTTGA -3´; 2nd PCR) and R- 7371 (5´- TTACAGTAGAAAAATTCCCC TCCACAATTAAA -3´; 2nd PCR). Cycling conditions for the second PCR were the same as for the first PCR except for an annealing temperature of 56˚C for 20 sec.

## Next generation sequencing

DNA quantification before NGS sequencing was done with the Quant-iT PicoGreen dsDNA Assay Kit (Invitrogen) according to the protocol. DNA concentrations were adjusted to 0.2 ng/μL and the Nextera XT DNA Library Preparation Kit (Illumina) was used according to the manufacturer's protocol. NGS was performed with an Illumina MiSeq Benchtop sequencer with 2x250bp reads. Next-generation sequencing data was processed with Seq-IT's in-house processing pipeline deepType-HIV. Upon iterative-mapping, reads covering the entire V3

loop were extracted and submitted to the geno2pheno[454] method. Ambiguous results were subjected to a secondary round of sequencing for confirmation.

## Sequence analysis

PCR amplicons were either sequenced by the Sanger sequencing (ABI 3730xl sequencer, Microsynth, Balgach, CH) or by NGS with an Illumina MiSeq Benchtop sequencer. Obtained sequences were assembled and aligned using Genious Prime2023.2.1 version. Larger sequence gaps from erroneous amplification of individual amplicons were corrected by incorporating the affected region from sequences that were identical outside the affected areas. Phylogenetic trees were constructed using the neighbor-joining method. Highlighter plots were compiled using an online tool (https://www.hiv.lanl.gov/content/sequence/HIGHLIGHT/highlighter_top.html.)

## Co-receptor usage prediction

We aligned the V3 amino acid sequences and predicted the co-receptor usage with the online analysis tool "geno2pheno" with false positive rate settings of 10% for CXCR4 detection. https://coreceptor.geno2pheno.org/index.php

## Statistical analysis

All data are presented as averaged values with median or mean ± SD; statistical significance was determined using unpaired student's t-test with two-tailed analysis or Two-way ANOVA was performed with Geisser-Greenhouse correction and Bonferroni's multiple comparison test; differences were considered statistically significant at $p < 0.05$. Analyses were generated with Prism (10.0.2, Graphpad Software, LLC).

## Supporting information

**S1 Fig. NGS analysis of the V3 region of viral RNA in plasma from clinical specimens. Related to Fig 1.** (A) P1. (B) P2. (C) P3. (D) P5. (E) P6. (F) P7. (G) P8. Pie charts depict HIV-1 RNA V3 loop diversity obtained by NGS. Sampling time points are shown above each pie chart (days during therapy). Variants with a frequency below 1% are combined in black, only the top 5 variants are depicted, each color representing one distinct virus variant. The total number of detected variants is summarized below each pie chart.
(TIFF)

**S2 Fig. Viral tropism dynamics over time. Related to Fig 1.** (A) P1. (B) P2. (C) P3. (D) P4. (E) P5. (F) P6. (G) P7. (H) P8. (I) P9. The frequency of R5- (cyan) and X4-tropism (pink) was determined by geno2pheno with an FPR cut-off of 10%. Only variants with a proportion ≥1% are shown. All individuals harbored B Subtype viruses.
(TIFF)

**S3 Fig. Proviral and poly-A transcript loads in VOAs. Related to Fig 2.** (A) Bar graphs of HIV poly-A loads / infected cells (HIV DNA) 5 days after stimulation. The black dotted line highlights the limit of detection. Corresponding individual IDs are above the respective bar plot. (B) Overall distribution of proviral DNA and cell-associated viral RNA per infected cell for each subset and all individuals. Black lines denote the mean value, and respective P values were determined by Two-way ANOVA and Bonferroni's multiple comparison test. Individual IDs are shown in legend next to scatter plots.
(TIFF)

**S4 Fig. Assessment of replication-competent virus. Related to Fig 2. VOA and GERDA readouts for tested individuals.** FACS plots show Gag vs. Env expression events in T cell subsets after *in vitro* T cell stimulation. Representative microscopic images show productive viral reactivation by LacZ (individual infection events in blue) after stimulation. (A) P1, (B) P3, (C) P4, (D) P6, (E) P7, (F) Uninfected PBMC as a negative control for GERDA and LacZ read-out of uninfected T cells post-stimulation (absence of blue cells). The scale bar is 250 μm.
(TIFF)

**S5 Fig. Phylogeny of 3' half SGA sequencing. Related to Fig 2.** Phylogenetic trees and highlighter plots of positive outgrowth wells. (A) SGA viral variants identified for P1, in TN and TCM stimulated cultures (TN well NO.1: blue, TN well No.3: red, TCM well No.3: green); B_1_17 was selected as the master sequence). Amplification gaps at the 5' end of B_1_C1, B_1_A7 or B_1_A14 were manually corrected, B_1_C11 was used to correct 5' region (red arrow). (B) Individual SGA variants for P7, with D_10 selected as a master sequence, D_B8 and D_A2 were manually corrected, D_B37 was used to correct 5' region (red arrow).
(TIFF)

**S6 Fig. T cell memory distribution. Related to Fig 2.** Distribution of all 4 sorted memory fractions. For each patient, ID and the respective age of the individual are given. % of each fraction is given in legend with respective memory assignment. The number next to the pie chart denotes the total amount of memory cells isolated for the respective patient.
(TIFF)

**S1 Table. Lab data of patient cohort.** NA: not available; ND: not detectable; [a] re-initiation of ART; [b] first ART initiation
(DOCX)

**S2 Table. Clinical data of patient cohort.**
(DOCX)

**S3 Table. V3 loop NGS data.** Representative NGS data of most detected proviral V3 variants of all individuals at the first sampling time point. Amino acid changes of individually aligned V3 variants are shown in black and underlined in bold.
(DOCX)

## Acknowledgments

We thank Lorena Urda (Molecular Virology, Basel), Rebekka Plattner, Kerstin Asal, and Louise Jayne Seiler (study nurses, University Hospital Basel) for excellent technical support, David Hauser and Christian Mittelholzer for critical review. We thank the participants of the SHCS for the provision of invaluable specimens.

## Author Contributions

**Conceptualization:** Yuepeng Zhang, Fabian Otte, Thomas Klimkait.

**Data curation:** Yuepeng Zhang, Fabian Otte, Alexander Thielen, Katharina Kusejko.

**Formal analysis:** Yuepeng Zhang, Fabian Otte, Alexander Thielen.

**Funding acquisition:** Yuepeng Zhang, Fabian Otte, Thomas Klimkait.

**Investigation:** Yuepeng Zhang, Fabian Otte, Thomas Klimkait.

**Methodology:** Yuepeng Zhang, Fabian Otte, Alexander Thielen.

**Project administration:** Yuepeng Zhang, Fabian Otte, Thomas Klimkait.

**Resources:** Marcel Stoeckle.

**Software:** Alexander Thielen.

**Supervision:** Thomas Klimkait.

**Validation:** Yuepeng Zhang, Fabian Otte, Alexander Thielen, Thomas Klimkait.

**Visualization:** Yuepeng Zhang, Fabian Otte.

**Writing – original draft:** Yuepeng Zhang, Fabian Otte, Thomas Klimkait.

**Writing – review & editing:** Yuepeng Zhang, Fabian Otte, Marcel Stoeckle, Alexander Thielen, Martin Däumer, Rolf Kaiser, Katharina Kusejko, Karin J. Metzner, Thomas Klimkait.

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
