## [Decision Letter · Decision Letter 0]

15 Apr 2024

Dear Prof. Klimkait,

Thank you very much for submitting your manuscript "HIV-1 diversity in viral reservoirs obtained from circulating T-cell subsets during early ART and beyond" for consideration at PLOS Pathogens. As with all papers reviewed by the journal, your manuscript was reviewed by members of the editorial board and by several independent reviewers. In light of the reviews (below this email), we would like to invite the resubmission of a significantly-revised version that takes into account the reviewers' comments. Please specifically address each concern of the reviewers and provide a significantly modified draft that provides more clarity of the findings and corrects any errors or mis-statements. 

We cannot make any decision about publication until we have seen the revised manuscript and your response to the reviewers' comments. Your revised manuscript is also likely to be sent to reviewers for further evaluation.

Sincerely,

Mary F Kearney

Guest Editor

PLOS Pathogens

Ronald Swanstrom

Section Editor

PLOS Pathogens

Michael Malim

Editor-in-Chief

PLOS Pathogens

orcid.org/0000-0002-7699-2064

Reviewer's Responses to Questions

**Part I - Summary**

Reviewer #1: The manuscript here by Zhang and Otte et al. examined Env V3 diversity in various T cell subsets during early ART to characterize the dynamics over time. Very interestingly, the authors found a strong association between the Torque-teno-virus (TTV) and levels of HIV-1 plasma viremia and viral transcription as measured by polyA. Additionally, the level of Env V3 diversity appeared to reflect TTV load and was dynamic pre-ART and following therapy. The authors bring attention to a possibility of utilizing TTV load for immune status as it relates to HIV infection.

While intriguing, there were several major concerns that should be addressed in order to strengthen the manuscript as there is a great amount of potential (detailed below). Further, please be mindful of Person First language by replacing instances of “patient”.

Reviewer #2: Zhang et al characterize HIV proviral (env) diversity between ART initiation and suppression in 8 participants (3 early infection, 2 chronic infection, 3 unknown), plus a ninth participant who was on long-term ART at study entry. The authors combine various techniques, including total proviral load measurements (single-target), within-host proviral diversity estimates obtained through bulk (NGS) sequencing of the V3 loop region, cell-associated poly-A RNA loads, along with single-genome sequencing (3' half) and viral outgrowth assays on sorted CD4+ T-cell subsets, to characterize the replication-competent HIV reservoir and overall proviral pool during this key phase.

This is an interesting question and valuable analysis, but the manuscript is unclear and confusing in parts, and contains what appear to be errors and mis-statements, which made it a challenge to read and understand. Also, the major stated conclusions of the study are not entirely novel (e.g. early ART limits HIV diversity, see PMID 24277811, 28046096) or potentially flawed due to analysis issues. Namely, the conclusion that replication-competent viruses were exclusively observed in TN or TCM is based on analysis of insufficient biological material, while the conclusions that poor immune status correlates with HIV diversity, and that TTV loads correlate with HIV transcription on ART, were interpreted from data that were incorrectly analyzed. While the conclusions may indeed be correct, as they are consistent with the reports of others, these issues represent major limitations.

Reviewer #3: This study aims at describing the HIV-1 reservoir diversity (based on diversity of the V3 loop) and infection competence in 9 individuals on ART. The main finding is that greater diversity is observed in TTMs and TEMs and that infection competent viruses are identified only in TCMs and TNs with low diversity. The study also shows inverse correlation between proviral diversity, and immune competence and a correlation between HIV and TTV viral loads.

The methods and analyses used are appropriate for addressing the research questions. A possible limitation would be the sole focus on the V3 region as representative of env diversity. The findings are an important addition to our understanding of HIV intrahost dynamics and evolution as they suggest a rudimentary difference between HIV diversity in different T cells.

Suggested minor revisions:

line 175: “Although in P5, plasma virus was fully suppressed by day 477 of therapy, intracellular HIV transcripts remained detectable at days 561 and 745 (Figs 1A and 1B).”

According to figure 1B there is no intracellular mRNA detected in P5.

Line 200: “For P1, we detected small proviral minorities in bulk CD4, TN, or TCM, e.g. on days 86 and 201 during therapy(Fig 1B) that were predominant with 90.4% of the virus population in TTM (Fig 2A). The same phenomenon was seen in the TTMs of P2, P3, P4, and P7, and in the TEMs obtained from P1, P2, P4, P5 and P6, (Fig 2A).”

This statement is somewhat inconsistent with the results. The first sentence is a little confusing, while it is true that there are small minorities in bulk CD4, TN and TCM, the time that these minorities were identified by NGS was according to the figure 2 legend, 5 days post-stimulation via QVOA which according to figure 1 was performed for P1 after day 400, therefore it is confusing that the authors continue with "e.g., on days 86 and 201" which were measured by NGS (presumably from bulk CD4+ T cells). Then the authors write that these minorities were predominant with 90.4% of the virus population in TTM (of P1) which is consistent with the figure, however in the following sentence they write that "The same phenomenon was seen in the TTMs of P2, P3, P4, and P7, and in the TEMs obtained from P1, P2, P4, P5 and P6", but this statement is inaccurate, because in these other participants the minority sequences do not reach 90% in TTMs in P2 and P3, and also they do not reach 90% in the TEMs of P1, P2, P5 and P6. To support a claim about increased diversity in TTMs and TEMs it would be beneficial to add a bar plot comparing the percentage of minority sequences in the different T cells, in each participant.

Line 209 (Figure 2 legend): “…B,C: NGS V3 loop diversity of viral RNA from QVOA positive wells. (B) Reactivated virus of TN and TCM form P1, C: Reactivated virus of TN form P7.”

For consistency, C should be in parenthesis.

line 224: “In the T-cell populations from 4/9 individuals (P4, P5, P9) no viral transcription could be induced(S3A Fig) and overall, levels of proviralHIV-1 DNA as well as viral transcripts were similar in every T cell subset(S3B Fig).”

Should be changed to 3/9 individuals.

In Fig S5A positions ~600 and ~1000 seem to be shared between the red and blue clades. This might be an indication of a recombination event in this region.

General comment:

Although the paper cites a couple of papers that also used V3 as a marker for diversity, it is not clear how representative the V3 diversity of other env regions? Is it possible that increased V3 diversity is the result of V3 targeting antibodies, and that in people with low V3 diversity other regions of env are targeted by antibodies and are therefore more diverse? It might be worth mentioning in the discussion.

**Part II – Major Issues: Key Experiments Required for Acceptance**

Reviewer #1: Please see full review.

Reviewer #2: I expand on the last two comments below:

1. It appears that the CD4+ T-cell subsets sorted for the QVOAs were obtained from 40ml of blood only. This is insufficient. It is therefore not surprising that viral outgrowth was only successful in 2 of 9 participants, where only Tn and Tcm yielded viral outgrowth. This is likely because these are typically the largest subsets, so possibly the only ones in which sampling was adequate to identify something. What were the bulk QVOA measurements for the study participants? The QVOA values obtained from Tn and Tcm subsets of the two successful participants were all < 1 IUPM, which further supports the need for much larger amount of biological sample to have been assayed.

2. The data in Figure 3 are analyzed incorrectly, as participants are represented multiple times on each plot. It looks as if as if the authors plotted all participants' longitudinal measurements as independent data points, and applied statistics, which is in violation of the i.i.d principle. There are dozens of data points in Figures 3A-C, for example, even though there are only 9 participants. Any analysis exploring correlations between participant-derived features (e.g. immune status and reservoir diversity) should represent each participant only once (or if it is a longitudinal analysis, with each participant measured once at each time point and the points connected). This could be done by, e.g. averaging the CD4 counts and diversity measurements for each participant. The same limitations apply to the analysis of TTV loads. This calls into question the validity of the conclusions made from all data in this figure.

Other comments

1. How was it determined that, of the 9 participants, 3 were diagnosed in primary and 2 in chronic infection (lines 105-107)? This needs to be described. The sentence prior (line 102-104) suggests that infection recency was predicted based on the participant proviral load measurement - but if this were the case, this would represent a major limitation (in addition, there may be an error in this sentence, which states that "a high proviral DNA load would be indicative of an early infection phase", which seems at odds with the evidence that early ART limits reservoir size).

2. Lines 145-147 state that proviral DNA loads and poly-A RNA levels were determined "at the time of diagnosis and at each follow-up visit" but this appears to be incorrect, as sampling from P1, P4 and P8 appear to be exclusively post-ART, and others appear to start on day of ART start. Please clarify.

3. It appears that, for the analysis of proviral DNA diversity in T-cell subsets presented in Figure 2, that the cells were cultured prior to analysis. Why was this done, and could this affect the distribution of variants observed?

4. Related to this, it appears that the data in Figure 2 were derived from a single time point, but this is not stated, and it is also unclear whether this is a different time point than those shown in Figure 1. Also, lines 198-199 state that "the viral reservoir remained largely stable in Tn and Tcm compartments compared to the bulk PBMC" but this is misleading as it suggests stability over time. But, the subsets analysis was cross-sectional.

5. Figure 2 interpretation: text line 203. The inclusion of P6 in this group is not supported by the data, as the variant is not "predominant" in Tem by any means

6. Figure S3 appears to be erroneous. S3B appears to plot the same data as S3A, but P5 had zero values for HIV polyA across all T-cell subsets in S3A, but has the highest polyA measurement for Ttm in S3B. Are the data mislabeled? Also please explain the letters after the subsets (S3A x-axis) and the numbers after the participants in the legend of 3B. Are the latter the time points sampled? Finally, consider plotting S3B data on a log scale, and using nonparametric stats as some data are non-normally distributed.

7. line 218-219 "Most integrated proviruses contribute to the generation of swarms of replication-defective genomes in the affected individual [39-40]". This wording is confusing and misleading. I believe what the authors mean is that the vast majority of proviruses that persist during ART are genomically defective.

8. line 231 (and elsewhere). Authors refer to their bulk amplification and NGS sequencing of the V3 region as simply "NGS analysis" but this can be confusing as sometimes the provirus is the target, and other times it's viral RNA. In the case of line 231 ("QVOA positive wells"), it is unclear which is being assayed. It would be helpful if authors could clarify the target throughout.

9. Figure S4. What were the criteria for determining the presence of replication competent HIV? It appears, for example that there are double-positive events in Ttm and Tem in participant 1. Also, please label the data for each participant on the figure itself, as they are not presented in numerical order.

10. Figure S5. How were QVOA wells chosen for sequence analysis? Were these the wells plated at (estimated) limiting dilution - in which case the observed diversity can be interpreted as diversity generated in vitro - or were these wells that likely contained more than one reservoir cell (or both)? The S5A highlighter plot does not strongly support the "close relationship" between Tcm and B_3 sequences suggested by the tree - can the authors include bootstrap values to give the reader an indication of confidence in the topology? Also it appears that there may be alignment issues for some sequences (e.g. B_1_C1 or B_1_A14)?

Reviewer #3: (No Response)

**Part III – Minor Issues: Editorial and Data Presentation Modifications**

Reviewer #1: Please see full review.

Reviewer #2: Minor comments

1. The first sentence of the last paragraph of the intro (lines 90-91) describes the study design and should go earlier e.g. before the summary of findings.

2. Figure S1 legend says the numbers under the pies are numbers of "detected variants" but this cannot be right. Are these total NGS read counts?

3. To respect current language guidelines, authors should use the term 'participant' or 'individual' throughout, and avoid the use of 'patient'

Reviewer #3: (No Response)

PLOS authors have the option to publish the peer review history of their article (what does this mean?). If published, this will include your full peer review and any attached files.

Reviewer #1: No

Reviewer #2: No

Reviewer #3: No
---

## [Decision Letter · Decision Letter 1]

21 Aug 2024

Dear Dr. Klimkait,

We are pleased to inform you that your manuscript 'HIV-1 diversity in viral reservoirs obtained from circulating T-cell subsets during early ART and beyond' has been provisionally accepted for publication in PLOS Pathogens.

Best regards,

Mary F Kearney

Academic Editor

PLOS Pathogens

Ronald Swanstrom

Section Editor

PLOS Pathogens

Michael Malim

Editor-in-Chief

PLOS Pathogens

orcid.org/0000-0002-7699-2064

Reviewer Comments (if any, and for reference):

Reviewer's Responses to Questions

**Part I - Summary**

Reviewer #1: Would like to thank the authors to thoroughly addressing all of this and other reviewer's comments.

Reviewer #2: (No Response)

Reviewer #3: The authors made all the requested changes except for one, they did not acknowledge a potential limitation of the study which is the use of V3 diversity as representative of HIV env diversity. It is this reviewer’s opinion that V3 is sometimes, but not always, representative of HIV env diversity. Therefore, the manuscript would be stronger if it included data to support the claim that V3 diversity is representative of total env diversity in this dataset. If not, it would seem reasonable simply to add a sentence acknowledging the potential limitation.

**Part II – Major Issues: Key Experiments Required for Acceptance**

Reviewer #1: Would like to thank the authors to thoroughly addressing all of this and other reviewer's comments.

Reviewer #2: (No Response)

Reviewer #3: (No Response)

**Part III – Minor Issues: Editorial and Data Presentation Modifications**

Reviewer #1: Would like to thank the authors to thoroughly addressing all of this and other reviewer's comments.

Reviewer #2: (No Response)

Reviewer #3: (No Response)

PLOS authors have the option to publish the peer review history of their article (what does this mean?). If published, this will include your full peer review and any attached files.

Reviewer #1: No

Reviewer #2: No

Reviewer #3: No

---

## [Editor Report · Acceptance letter]

31 Aug 2024

Dear Prof. Klimkait,

We are delighted to inform you that your manuscript, "HIV-1 diversity in viral reservoirs obtained from circulating T-cell subsets during early ART and beyond," has been formally accepted for publication in PLOS Pathogens.

Best regards,

Michael Malim

Editor-in-Chief

PLOS Pathogens

orcid.org/0000-0002-7699-2064